# Investigating the Impact of Feature Reduction for Deep Learning-based Seasonal Sea Ice Forecasting

Lars Uebbing*[1], Harald L. Joakimsen[1], Luigi T. Luppino[1], Iver Martinsen[1], Andrew McDonald[2,3], Kristoffer K. Wickstrøm[1], Sébastien Lefèvre[1,4], Arnt-Børre Salberg[5], J. Scott Hosking[3,6], and Robert Jenssen[1,7]

[1]UiT The Arctic University of Norway
[2]University of Cambridge
[3]British Antarctic Survey
[4]University of Southern Brittany
[5]Norwegian Computing Center
[6]The Alan Turing Institute
[7]University of Copenhagen
{lars.uebbing, harald.l.joakimsen, luigi.t.luppino, iver.martinsen, kristoffer.k.wickstrom, robert.jenssen}@uit.no
{anddon76, jask}@bas.ac.uk
sebastien.lefevre@irisa.fr
salberg@nr.no

## Abstract

With the state-of-the-art IceNet model, deep learning has contributed to an important aspect of climate research by leveraging a range of climate inputs to provide accurate forecasts of Arctic sea ice concentration (SIC). The deep learning subfield of eXplainable AI (XAI) has gained enormous attention in order to gauge feature importance of neural networks, for instance by leveraging network gradients. In recent work, an XAI study of the IceNet was conducted, using gradient saliency maps to interrogate its feature importance. A majority of XAI studies provide information about feature importance as revealed by the XAI method, but rarely provide thorough analysis of effects from reducing the number of input variables. In this paper, we train versions of the IceNet with drastically reduced numbers of input features according to results of XAI and investigate the effects on the sea ice predictions, on average and with respect to specific events. Our results provide evidence that the model generally performs better when less features are used, but in case of anomalous events, a larger number of features is beneficial. We believe our thorough study of the IceNet in terms of feature importance revealed by XAI may give inspiration for other deep learning-based problem scenarios and application domains.

## 1   Introduction

Arctic sea ice plays a pivotal role in our earth's climate system [1]. In recent years, drastic shrinkage of the sea ice extent has been observed due to anthropogenic climate change [2]. This development is

*Corresponding Author.

particularly worrying as a reduction in sea ice again accelerates global warming [3]. Accurate forecasts of seasonal sea ice help our general understanding of the earth's climate but can also be put to use directly, e.g. to estimate possible shipping routes that depend on the extent of sea ice.

Recently, Andersson et al. introduced the deep learning model IceNet that forecasts average sea ice concentration (SIC) with high accuracy for lead times up to 6 months [4]. Long lead times are particularly challenging due to the spring predictability barrier [5], which is why other models are often restricted to short-term predictions [6–8]. IceNet uses a whole range of different climate observables as input features and provides very accurate forecasts, in particular for anomalous events.However, the predictions are not easy to interpret and the question was posed from which features the network draws the information that leads to its accurate forecasts.

Joakimsen et al. [9] leverage a gradient based method to provide an extensive deep learning XAI [10] analysis of the IceNet's feature importance. Thereby, they focus on the forecast for the anomalous month September 2013, as the IceNet showed a particularly high accuracy in this prediction.The results yield detailed information about the impact of the individual features with spatial resolution and with respect to lead times. Based on their results, Joakimsen et al. conclude that only a fraction of the input features provide a relevant contribution to the forecast and suggest that a model trained with only a few features should maintain a high accuracy.

Convolutional neural networks are computationally demanding and typically require substantial storage capacity [11]. There has been a lot of effort to leverage feature importance scores to prune parameters and reduce redundancy, as this offers a

Proceedings of the 6th Northern Lights Deep Learning Conference (NLDL), PMLR 265, 2025.

way to reduce storage requirements and computation costs while maintaining a high accuracy [12]. In contrast to previous studies that often cut back individual connections, node, etc. [13], we want to examine a more radical approach by completely discarding the features with low importance scores. This has a distinct advantage because it entirely removes the need for a portion of the input features, that in many cases might be hard to come by. Inspired by the findings of Joakimsen et al. [9], we conduct a novel analysis where we train model variations of the IceNet with different configurations of input features. We compare the performance for the different configurations for the case that was studied by Joakimsen et al. in detail and investigate how the results generalize for all predictions. Finally, we separate a set of anomalous events to examine how the models compare when it comes to predict outliers.

## 2 Related Work

Here, we present the work of Andersson et al., which introduces the IceNet model, as well as the work of Joakimsen et al., that interrogates IceNet's feature importance.

*A. IceNet*
In 2021, Andersson et al.'s work on the IceNet was published. It shows remarkable accuracy for the prediction of SIC, in particular when it comes to extreme events and long range forecasts. In its original form, the IceNet takes 50 input features, which comprise of: SIC observations from the preceding 12 months, a linear trend forecast (LTF) of the SIC for the next 6 months, 11 climate variables (1-3 months prior), seasonal encodings and meta data (land masks). Each of the features is spatially represented by a $432 \times 432$, image-like data frame, whereas each grid cell or pixel corresponds to a $25 \times 25$ km area in the northern hemisphere. The LTF is calculated by taking the previous 35 years of SIC data for each month and pixel individually and produce a linear fit through these points. That means, the LTF of an individual pixel is calculated based on the SIC values that were obtained for the same pixel in the same month within the previous 35 years. The best linear fit through these values produces the LTF for the same month in the subsequent year.

The model itself is a convolutional neural network with a U-Net [14] architecture (see Figure 1). The sea ice prediction is arranged as a classification problem with the 3 SIC classes

1. **open-water:** SIC $\leq 15$ %

2. **marginal ice:** 15 % < SIC < 80 %

3. **full ice:** SIC $\geq 80$ %.

The model is trained to forecast probabilities for the individual grid cells to fall into any of these classes. Thus, the prediction for any month consists of three $432 \times 432$ maps of probabilities, one for each SIC class. In this manner, the model directly produces forecasts for lead times of 1 to 6 months for any given initialization month.

To increase the robustness, Andersson et al. train an ensemble of 25 models like this, using different random initializations. The mean of the individual predictions yields the finial forecast.

A transfer learning approach is used to train the model. First, the model is pretrained on climate simulation data (CMIP6) from 1850 to 2100. Then, the training is continued on monthly averaged observation data (era5) from 1980 to 2012. Detailed information about the type, origin and preprocessing of the data can be found in [4] and on GitHub[1].

*B. Interrogating Feature Importance*
Triggered by the accurate forecasts of IceNet for extreme events, Joakimsen et al. published an XAI study with the aim to identify the features, that are most relevant for these results.

There are several approaches on how to estimate feature importance for a deep neural network [15–17]. Gradient based saliency maps [18], as they are used in by Joakimsen et al. [9], offer a way to not only assign importance scores to the individual features, but also provide information on whether or not features have a positive or negative impact on the predictions. Furthermore, this method is spatially resolved, which is particularly useful when there are regions of special interest in the forecasts.

The gradient of a function can be seen as a measure of its sensitivity with respect to small changes of the input variables. Let $\mathbf{x} = \{x_1, ..., x_K\}$ be a set of $K$ input features that result in a prediction $f(\mathbf{x})_{mn}$ for the grid cell $(m, n)$, with $m \in \{1, ..., M\}$ and $n \in \{1, ..., N\}$, whereas $M$ and $N$ are the number of rows and columns of the grid. A gradient saliency map can be created with respect to a distinct feature $x_k$, by computing the gradients of the prediction $f(\mathbf{x})_{mn}$ with respect to all spatial components $x_k(i, j)$ of the input feature $x_k$ and accumulating over the spatial components $(m, n)$ of the prediction as follows:

$$R(x_k(i,j)) = \frac{\partial}{\partial x_k(i,j)} \left( \sum_{m=1}^{M} \sum_{n=1}^{N} f(\boldsymbol{x})_{mn} \right). \quad (1)$$

The value of $R(x_k(i,j))$ yields information on how a change of the $(i, j)$-component of feature $x_k$ influences the overall prediction. In order to get a single value $R(x_k)$ to rank the feature importance, it is

[1] https://github.com/tom-andersson/icenet-paper

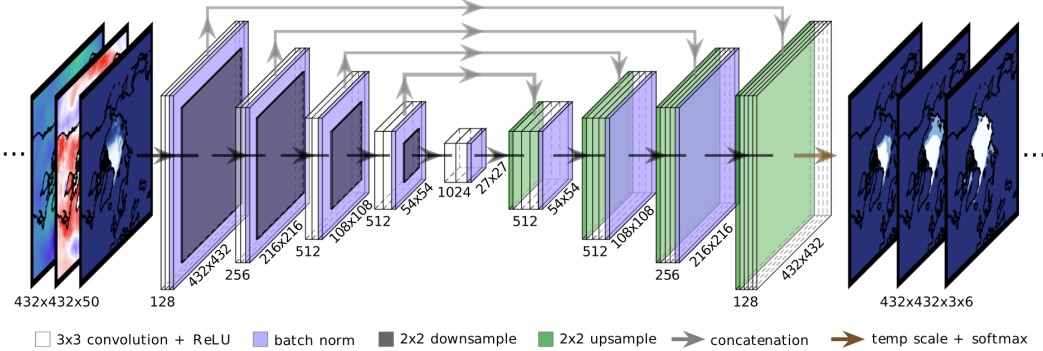

**Figure 1.** IceNet's U-Net architecture takes a stack of $432 \times 432$ input features and processes them with an encoder-decoder structure to output 6 months of forecast, each separated into 3 SIC classes. Image taken from [4].

summed over the spatial components $(i, j)$:

$$R(x_k) = \sum_i^M \sum_j^N R(x_k(i,j)). \qquad (2)$$

Joakimsen et al. use this method but sum only over a specific region of interest, that corresponds to the area of unusual sea ice extent. This way the result is more meaningful with respect to the anomalous part of the forecast. Focusing on this application on the particular anomalous month September 2013, they provide results that suggest only few of the 50 input features are important for the forecast of the anomalous sea ice extent, namely the historic SIC, the LTF, seasonal encoding and the land masks. They conclude that IceNet should still yield accurate forecasts, when only these input features are considered. [9]

We acknowledge that there is an ongoing discussion about the reliability and trustworthiness of the results from gradient-based XAI methods [19, 20]. Future works will therefore aim to investigate alternative XAI methods [21–23] to see if similar conclusions as in Joakimsen et al. are reached.

## 3 Methodology

Based on the importance scores provided by Joakimsen et al., we want to investigate how the IceNet model performs, when features with low importance are discarded. In this section we present our changes to the original model and our approach to evaluate the generalization of the results.

### A. Feature Reduction and Retraining
To test how IceNet performs under reduction of input features, we set up different feature configurations:

1. **original:** This configuration contains all 50 features that were used in the original IceNet by Andersson et al.
(total features: 50)

2. **reduced:** This configuration discards all 11 climate variables but contains all 12 SIC observations, the LTF, seasonal encodings and meta data (land masks).
(total features: 21)

3. **minimal:** This configuration only contains the LTF, seasonal encodings, meta data (land masks) and one SIC observation of the preceding month.
(total features: 10)

The *reduced* configuration includes the features that Joakimsen et al. suggested to be sufficient for a good forecast, while the *minimal* configuration represents a further shrunk set of features that sets a higher threshold for the importance scores of a feature to be included. For each of these configurations we train an ensemble of 10 models with different random initializations but the same architecture. We do not pretrain the models on simulation data, as it is computationally expensive and it was shown that the benefit particularly for the critical months is very little [4]. Instead the models are trained purely on monthly observational data from 1980 to 2011. The data of 2012 - 2017 is assigned for validation and a test set contains the data from 2018 - 2020.

### B. Performance Evaluation
Consistent with [4], the performance of the trained model is evaluated using a binary accuracy measure, based on the 15 % threshold, which is a common metric to measure differences in sea ice extent [24]. Each cell is regarded as either *ice* (SIC > 15 %) or *no ice* (SIC ≤ 15 %). As for the predictions, that means if the accumulated probability of the classes 2 (marginal ice) and 3 (full ice) is above 50 % the cell is regarded as *ice*, otherwise it is considered to be *no ice*. The binary accuracy calculates as the percentage of correctly classified grid cells for every individual prediction. In addition to the pure measure of accuracy, we use the standard deviation between ensemble members to provide a brief uncertainty estimation for the different IceNet con-

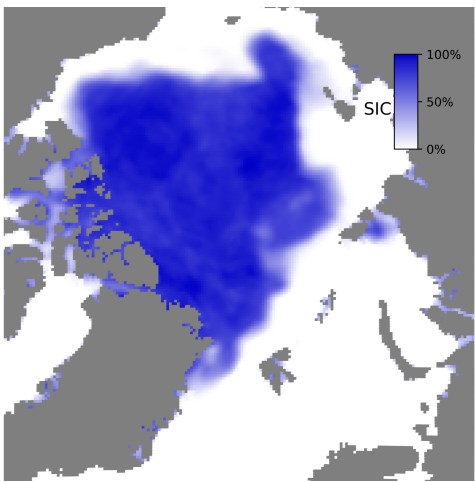

**Figure 2.** Anomalously large SIC (%) during September 2013 in the northern hemisphere.

figuration in Appendix A.

To get a deeper understanding of how the feature reduction affects the predictions beyond simply measuring the average accuracy, we look at different cases separately. The analysis of feature importance by Joakimsen et al. was performed on the prediction for September 2013, as this was a particularly anomalous but accurately predicted event. Thus, we will first compare how the reduced input features affect the model predictions for this particular month in detail. Next, we look at the general case, where we include all predictions to see if the results that were obtained for September 2013 generalize. Last we separate a set of predictions for that we classify as anomalous months and compare the model performances for these predictions. With this set of experiments we aim to analyze the impact of feature reduction on general predictions but also to uncover how anomalous events relate to that, as they are of particular interest. Further, we can put the prediction for September 2013 into context and use it to reveal some details of how the different predictions differ.

## 4 Results

In this section we provide the results of three experiments and evaluate the results with respect to the impact of feature reduction in different scenarios.

### A. September 2013 in Detail

Figure 2 shows the observed SIC for the anomalous September 2013. In particular the upper right region represents an unusual extent of sea ice [9]. In contrast, Figure 3 provides the deviations of the individual model predictions from the observation for a lead time of one month. A supplementary figure, showing the results also for a lead time of 6 months can be found in Appendix B, Figure B.1. The pix-

els are color-coded, with red areas corresponding to pixels where sea ice was observed but not predicted, and blue areas for pixels where sea ice was predicted but not observed. We can clearly see that all (mis-)predictions have the same overall structure, with false predictions located around the borders of the ice surface. The tendencies of predicting too high or too low SIC are distributed very similarly, with generally too much sea ice in the regions north of Europe extending to mid Russia and too little sea ice north of Canada, Alaska and eastern Russia. Considering that in this month, an anomalous large extent of sea ice has been observed, it is surprising that none of the models seems to predict generally too little sea ice. Instead, it seems like the whole sea ice surface of the predictions is shifted towards Europe compared to the observed sea ice. Sea ice drifts are mainly determined by wind [25]. The original IceNet configuration is the only one that takes wind as input feature but the results show that this model could not predict this shift of the sea ice surface any better than the models without wind.

Another key observation concerns an area in the right center of the plots (circled by a dashed line in Figure 3(a)) which contained ice at the targeted time. Both of the reduced models could predict this area very accurately for a lead time of one month, while the original IceNet was not able to pick up on indications for this.

Figure 4 shows the binary accuracies for the predictions of all models for lead times from 1 to 6 months. Supplementary figures of the binary accuracies which include uncertainty estimations can be found in Appendix A. The accuracies between the models for a given lead time vary slightly but remain in the same domain and thus, support the results of Figure 3. It is notable that the binary accuracies of the reduced models both exceed the accuracy of the original model with 1.2 and 1.0 percentage points (pp.) for a lead time of 1 month. These results strongly support the hypothesis that IceNet's good performance for the prediction of the extreme event in September 2013 is mainly based on previous SICs. Also the observation that the accuracy of the reduced models increases relative to the original model matches the results of Joakimsen et al.

### B. Overall Performance

Next we examine whether this behavior also extends to the general model performance, apart from this individual extreme event. For this purpose, we average the binary accuracies for each lead time over all predictions from 2012 to 2020. Figure 5 shows the resulting average accuracy versus lead time for each configuration. The original IceNet configuration yields a slightly (ca. 0.5 pp.) lower accuracy than observed by Andersson et al., but this can be

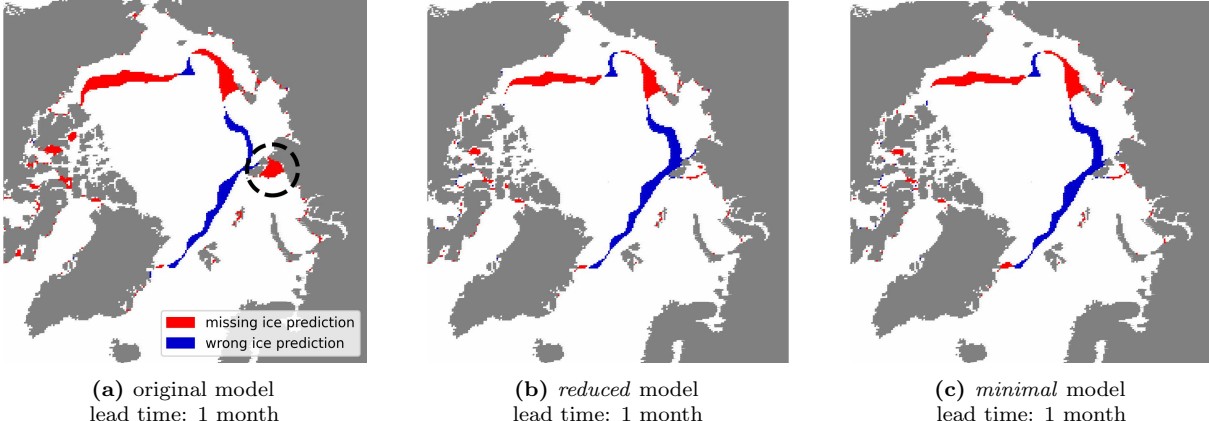

**(a)** original model
lead time: 1 month

**(b)** *reduced* model
lead time: 1 month

**(c)** *minimal* model
lead time: 1 month

**Figure 3.** Deviations of the binary IceNet predictions from observed data for September 2013 for a lead time of one month. Blue areas correspond to false positive predictions and red ares to false negative predictions, respectively. The individual plots represent the results for the different IceNet configurations.

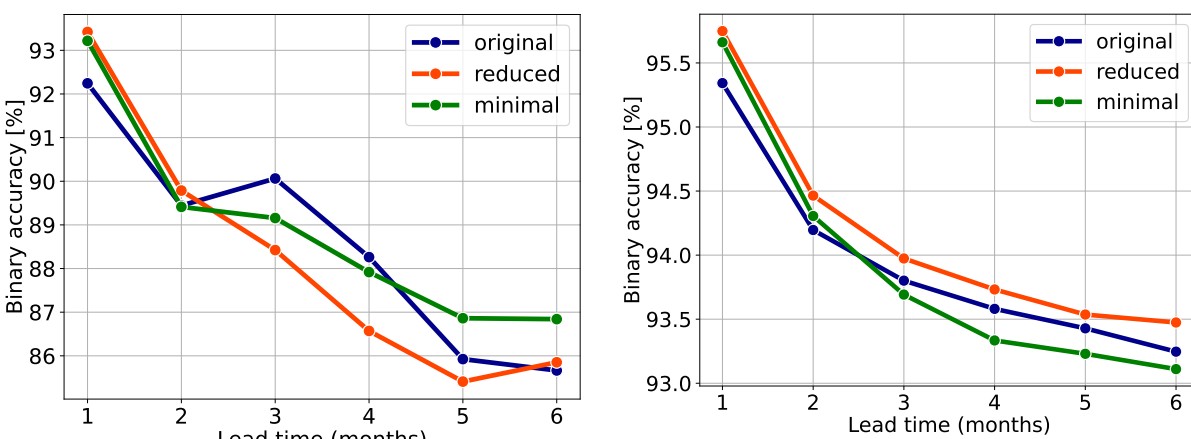

**Figure 4.** Average binary accuracy of the three different IceNet configurations plotted versus lead time for September 2013.

**Figure 5.** Average binary accuracy of the three different IceNet configurations plotted versus lead time.

explained by discarding the pre-training and the lower number of 10 ensemble members in our experiment compared to 25 members in the experiments of Andersson et al. Remarkably, while decreasing in the same manner, the *reduced* IceNet is 0.1 - 0.4 pp. more accurate over all lead times, with a maximum accuracy of 95.7 % for 1 month lead time. Even the *minimal* configuration of the model shows higher accuracy than the original version for lead times up to 2 months. From lead times of 3 months and up, the accuracy drops below the original one. While nearly matching the accuracy of the *reduced* model for small lead times, the *minimal* model's accuracy is clearly the lowest for large lead times and thus, decreases faster with increasing lead time. These results show that Joakimsen et al.'s hypothesis, which corresponds to the *reduced* model, holds true even for the general case. For longer lead times it seems that not only the LTF is relevant, but also the monthly SICs of the preceding year, as the *min-*

*imal* model's accuracy decreases quicker compared to the configurations that include these SICs. It should be noted that the LTF itself already provides a good estimate for the future SIC [4], particularly for non-extreme events. Thus, being able to predict anomalous events with a high accuracy holds more value than regular predictions.

*C. Performance for Anomalous Months*

So far, we just analyzed the model performances in general and for one particular extreme event. In the next step, we therefore examine how the different models compare for cases that we classify as anomalous, without focusing on one explicit event. As a metric to determine how anomalous an event is, we use the binary accuracy of the LTF. If the LFT has a high accuracy, it means the SIC for the given month is very similar to the expectation based on the SIC of the previous years. A low LTF accuracy can thus be interpreted as an anomaly. To show how the different IceNet configurations behave with

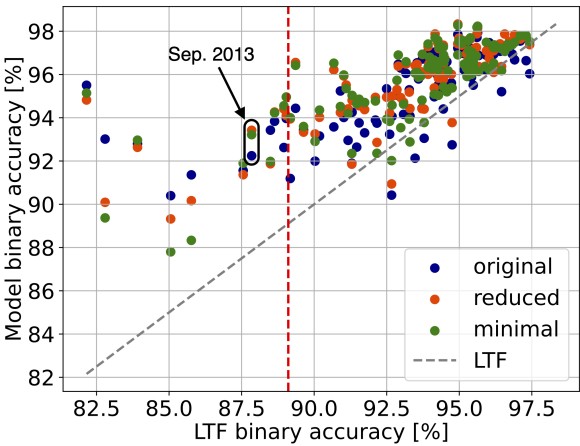

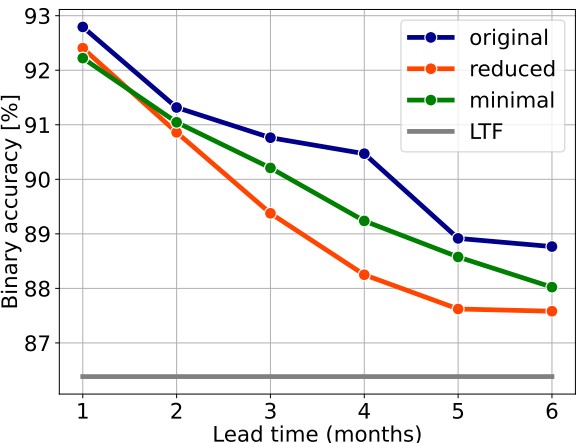

**Figure 6.** Binary accuracies of individual IceNet predictions (lead time of 1 month) with different feature configurations plotted versus the binary accuracy of the LTF. The dashed grey line marks the accuracy of the LTF as a reference. The points left of the dashed red line indicates the border correspond to the 10 % most anomalous events.

**Figure 7.** Average binary accuracy of the three IceNet configurations for the 10 % most anomalous events plotted versus lead time.

## 5 Conclusion

We have trained versions of the IceNet model using different configurations (original, *reduced*, *minimal*) of input features. For these models we provided an extensive performance analysis including different sets of predictions. Our results show that averaged over all predictions, the *reduced* model yields the highest accuracy for all lead times. The *minimal* model shows an increased accuracy for lead times up to two months but drops below the original model for larger lead times. For the particular event of September 2013, we also demonstrated that the reduced versions capture properties of the ice structure, that the original version missed. In the end we show that the original model remains superior in cases that deviate a lot from the usual SIC for a given month.

respect to the grade of anomaly, Figure 6 shows the binary accuracies of the IceNet forecasts with a lead time of 1 month plotted versus the accuracy of the LTF. The figure shows that IceNet's accuracies are generally lower when also the LTF accuracy is low. But at the same time the accuracies distinguish more from the LTF line, for low LTF accuracies. In other words, the more the observed sea ice deviates from its usual extend for a given month, the more superior are the IceNet predictions compared to the LTF. While this view makes it hard to draw general conclusions about the differences between the IceNet configurations, the figure shows that for most extreme events the original configuration performs better than the reduced versions and that the predictions for September 2013 (marked in the figure) is just an exception.

To evaluate the performance for extreme events more quantitatively, we classify the 10 % lowest LTF accuracies as anomalous / extreme and assess the performance for these months separately. That corresponds to the predictions left of the red dotted line in Figure 6. Figure 7 shows the average accuracy for these extreme events versus the lead time for all the configurations. Compared to Figure 5 we can see that the ranking has changed and in fact, the original IceNet has the highest accuracies for all lead months. While all models have a similar accuracy of 92 - 93% for a lead time of 1 month, their difference increases with lead time up to about 1.2 pp. between the *reduced* and original model for a lead time of 6 months. We can also see that the accuracy drop from lead time of 1 month to 6 months is more significant (~ 4 - 5 pp.) than in the general case (~ 2 pp.) for all models.

We conclude that XAI studies as provided by Joakimsen et al. [9] can be leveraged to effectively minimize the amount of input features for deep learning models, by maintaining overall high accuracy, or even increasing it. This yields a practical and straightforward method, e.g. for cases when certain data is not easily obtainable or data storage is an issue. For the generalization to extreme events and outliers, however, models might still benefit from additional features.

Future work might investigate the computational benefits of decreasing the number of features. Further studies might benefit from more extensive underlying XAI studies that, e.g. include different methods to estimate feature importance to increase reliability. Additionally, the robustness of the models might be analyzed by introducing perturbations to the model. Interesting insights could also be gained by going deeper into the uncertainty estimation, for example by training several ensembles per configuration and compare the accuracy deviations of the ensembles within one configuration.

# Acknowledgments

This work was partially funded by the Research Council of Norway (Visual Intelligence, grant no. 309439 and KnowEarth, grant no. 337481).

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

# A   Uncertainty Estimation

In section 4, we discussed the performance of different IceNet configurations in terms of the binary accuracy, using the predictions given by an ensemble of 10 models per configuration. Here, we leverage the standard deviation of the ensemble members to give a simple estimate for the uncertainty of the results.

Each ensemble member yields individual predictions and thus, an individual accuracy score per predicted month and lead time. In order to supplement the our performance analysis in a meaningful way, we want to leverage the standard deviation of the ensemble members to give a simple estimate for the uncertainty of the results. Each ensemble member yields individual predictions and thus, an individual accuracy score per predicted month and lead time. According to our performance analysis, the calculation of the standard deviation should be performed in such a way that we get distinct results per lead time and set of predictions. We could calculate the standard deviation between the individual model accuracies, taking into account all of their predictions for a set of dates and fixed lead time at once. However, to reduce the effect of the size of the data set, i.e. the number of dates included into the calculation, we decide to calculate the standard deviation of an ensemble for each lead time and each prediction at a time. Thus, for each ensemble we get one value per prediction month and lead time. For the evaluation of a prediction set, we average over the respective standard deviations. To show the results of our uncertainty estimation, we reproduce Figure 4, Figure 5 and Figure 7, which show the accuracies for different sets of predictions and we add the respective standard deviations as error bars. These plots are shown in Figure A.1, Figure A.2 and Figure A.3, respectively.

For the prediction of September 2013 (Figure A.1), the standard deviations differ a lot between model configuration and lead times. This is can be at-

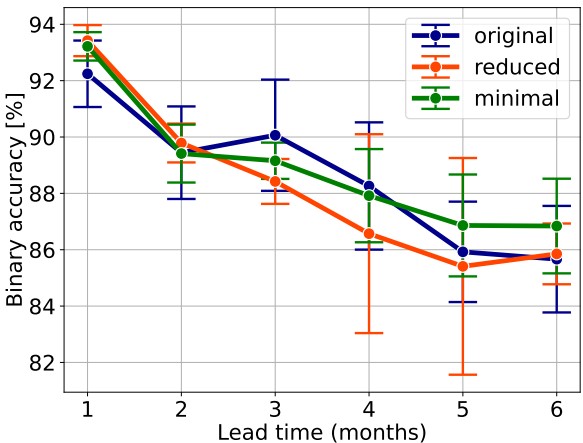

**Figure A.1.** Average binary accuracy of the three different IceNet configurations plotted versus lead time for September 2013. The plot shows the accuracy standard deviation of the ensemble members for this prediction as error bars.

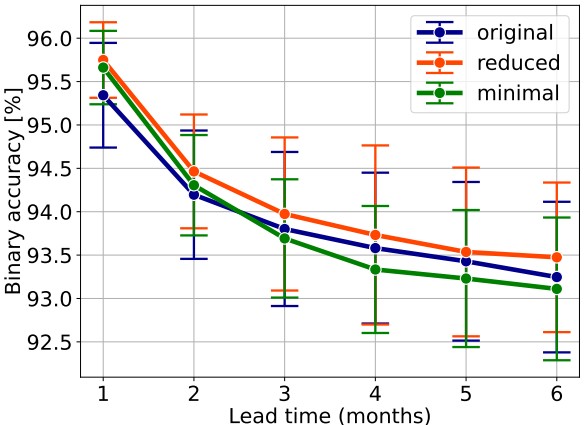

**Figure A.2.** Average binary accuracy of the three different IceNet configurations plotted versus lead time. The plot shows the average accuracy standard deviation of the ensemble members for the respective predictions as error bars.

tributed to the fact that we are only looking at a single prediction and individual differences contribute a lot to the standard deviation.

Figure A.2 and Figure A.3, showing the corresponding plots for all available predictions and the 10 % most anomalous months, respectively, are show more consistent standard deviations. Overall, both plots show that all three IceNet configurations tend to increase in their uncertainty as the lead time in creases. A comparison of both figures shows that the uncertainty for the anomalous events is in most cases larger than for the general case that includes all predictions. However, this effect might be enhanced by the fact that the number of predictions included for the anomalous events is much smaller and thus, individual fluctuations have a larger impact.

Even though, e.g. for the lead time of 4 months

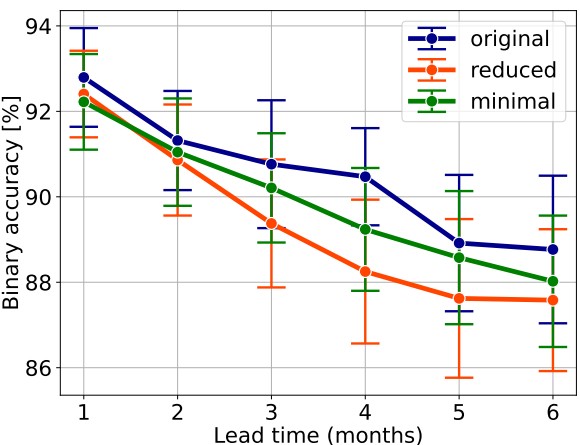

**Figure A.3.** Average binary accuracy of the three IceNet configurations for the 10 % most anomalous events plotted versus lead time. The plot shows the average accuracy standard deviation of the ensemble members for the respective predictions as error bars.

in the general case (Figure A.2), the standard deviation of the *reduced* configuration is clearly larger than the one of the original configuration, it can be stated that overall the uncertainty of all three configurations are in a similar regime and there are no distinct differences. It should also be noted that the standard deviations are generally very large and in most cases exceed the differences between the different averaged accuracies of the three configurations. This indicates that the differences observed between the configurations might not be as significant. However, more sophisticated and detailed analyses are necessary to give reliable results and interpretations of the model uncertainties.

# B Prediction Deviations for September 2013

Figure B.1 shows the deviations of the September 2013 forecasts for the three IceNet configurations. Areas in red show regions where the models falsely predicted no ice and areas in blue correspond to regions where the models falsely predicted ice. This figure extends Figure 3 from section 4 by adding the forecasts with a lead time of six months to the one month forecasts. It shows, that for longer lead times, i.e. predictions of this months further ahead of time, all models mispredicted the region in the right center which is highlighted in Figure B.1(a).

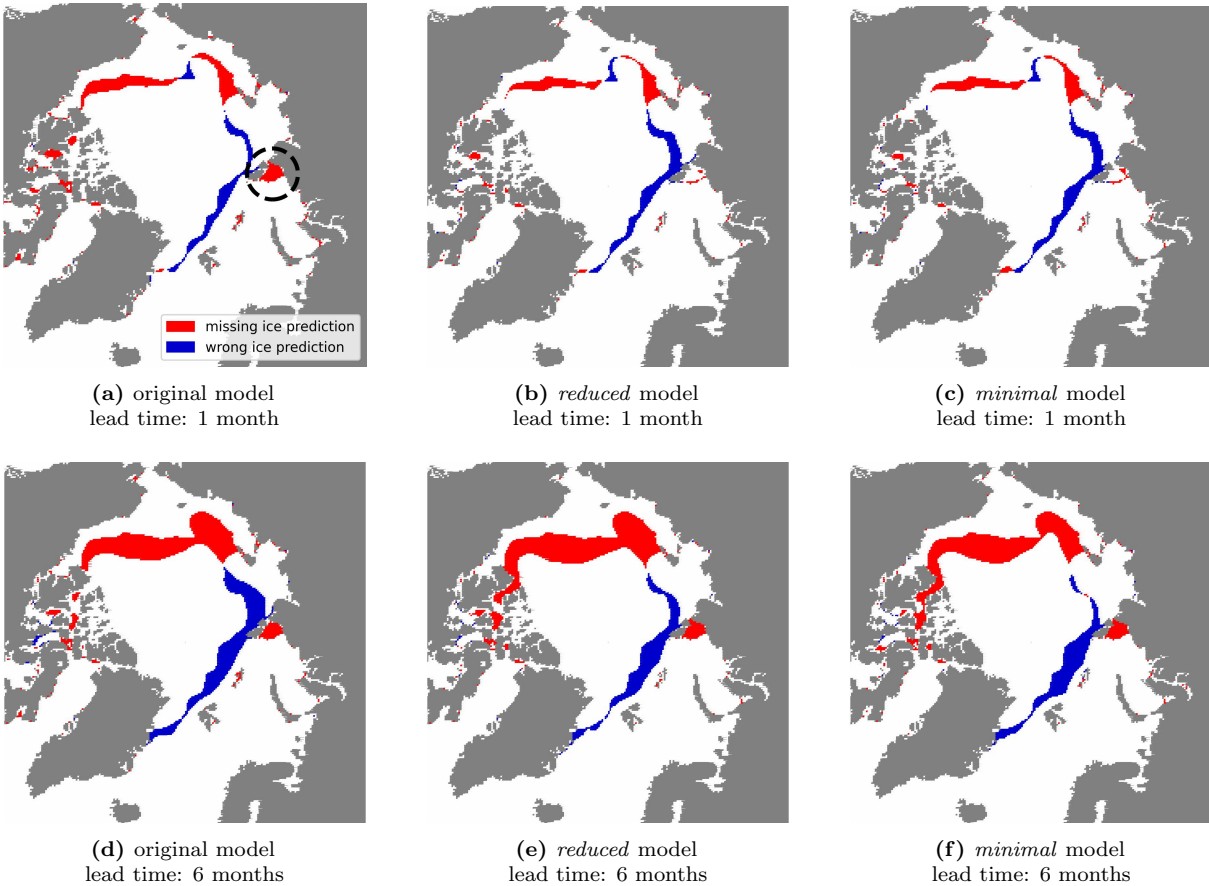

**(a)** original model
lead time: 1 month

**(b)** *reduced* model
lead time: 1 month

**(c)** *minimal* model
lead time: 1 month

**(d)** original model
lead time: 6 months

**(e)** *reduced* model
lead time: 6 months

**(f)** *minimal* model
lead time: 6 months

**Figure B.1.** Deviations of the binary IceNet predictions from observed data for September 2013. Blue areas correspond to false positive predictions and red ares to false negative predictions, respectively. The upper row ((a) - (c)) corresponds to predictions with a lead time of 1 month and the lower row ((d) - (f)) to predictions with a lead time of 6 months. The individual plots in each row represent the results for the different IceNet configurations.

