# OpenReview forum: "Investigating the Impact of Feature Reduction for Deep Learning-based Seasonal Sea Ice Forecasting"
_NLDL.org/2025/Conference — NLDL 2025 Oral_

### Official Review · Reviewer_9b9J · 2024-10-04
**Sound manuscript with valuable evaluations and results for the machine learning sea ice coverage prediction audience**

**Confidence:** 4

**Summary:**

The manuscript presents a study that expands on the previously published machine learning model IceNet for sea ice concentration forecasts ranging from one through six months lead time in the northern hemisphere. Results suggest that a meaningful reduction of the number of input features (based on their importance score) can improve the forecast quality under normal conditions. In extreme situations and anomalies, though, the full set of input features results in an improved forecasts of sea ice cover compared to when using a reduced or minimal set of input features.

**Strengths:**

### Reliability
Multiple seeds ensure the reproducibility and reliability of the results. In the figures, though, it would be great to see the standard deviation of the 10 different models (as error bars or as shaded area surrounding the mean line).

### Soundness and Approachability
The manuscript is sound and clearly written. It outlines well the research goals and how they are obtained. Results are well underlined with numerous plots, which are interpreted concisely.

**Weaknesses:**

### Structure
The manuscript would benefit from a structural rework. For example, the transition from Methodology to Results appears somewhat abrupt and the Methods section seems more like Related Work or Foundations. I'd thus suggest to rename section 2 into Related Work and Methodology (or to create two separate sections) and outline the manipulations or modifications this manuscript adds to prior work (such as currently contained at the beginning of the Results section). Also, the Methods section could be subsectioned to outline the conceptualizations of the two core aspects of this work: (a) retraining IceNet with a reduced set of features, and (b) investigating IceNet's generalization performance. The Results section could be subsectioned accordingly.

### Clarity
- Can authors please include information about how IceNet generates predictions out to 6 months? That is, does it run autoregressively, are separate models trained for each lead time, or does the model generate all outputs at once?
- Also, it is unclear how the Linear Trend Forecast models is designed and where the data comes from. In this vein, lines 104-105 could be extended to contain concrete information about the satellites that recored the data, where it is available, and how it has been preprocesses.
- Figures could benefit uppon inclusion of a grid. Once done, concrete values can be extracted from the Figures and Table 1 could be converted into a figure similar to Figure 4 to improve the presentation of results and make it more coherent.
- Does Figure 5 contain only extreme events (as suggested in lines 298-302)? This might be emphasized around lines 286-288 and also added to the figure caption. If Figure 5 does not only contain extreme events, can the authors mark extreme events explicitly?


### Minor remarks and typos
- Line 143 remove one "the"
- In Line 167, add "(land masks)" to the text again to repeat what is meant with metadata.
- When talking about less ensemble members compared to Andersson et al., please add the number of ensembles used there.
- Missing "s" at "drop" in line 316?

**Final Rebuttal Confidence:**

4

**Final Rebuttal Justification:**

The authors carefully addressed and solved my concerns. During the rebuttal, one downside became clear, namely a large overlap of standard deviations for different configurations. This weakens the relevance of different input configurations. Nevertheless, the analyses and results are worth sharing with the community.

**Justification:**

Even though the manuscript can be improved in various ways, it contains valuable evaluations that are worth sharing with the ML and sea ice research community. In particular, this research reveals the value of considering a full set of input variables when aiming for capturing anomalous events with higher accuracy. To my understanding, these results differ from previous work which suggested the use of a minimal set of input variables, even for extreme events. The manuscript presented here, though, reveals that the conclusions in the former study might wrongly been drawn from a single example. When repeating the analysis with a statisticially meaningful number of extreme events, the pattern reverts and suggests the employment of many input features.

My recommendations for revisions do not ask for additional analyses but ask for more information and for a reordering of the section, which I consider solvable in the rebuttal period. Thus, I recommend to accept this article for the conference.

---

> ### Author Rebuttal · Authors · 2024-10-25
>
> We want to thank the reviewer for his time to prepare this very helpful and structured comment.
>
>
> We prepared figures that include the standard deviations. For better visibility, we kept the original figures in the main text and put the figures with standard deviations, as well as their discussion in the appendix.
>
>
> Structure:
>
> In the revised paper, we adjusted the structure by adding a “Related Work” section and by introducing subsections according to the reviewers comment.
>
>
> Clarity:
>
> The 6 months predictions are produced at once. We state this more clearly in the revised paper.
> We added a more extensive explanation of the LTF in the “Related Work” section
> Due to a lack of space, we did not go into detail regarding the kind of data, its preprocessing, etc. but in the revised paper we point to where the information can be found.
> We added a marking for the extreme events in the respective figure.
>
>
> Minor remarks:
>
> We adopted all suggestions of the reviewer.

---

### Official Review · Reviewer_HRXG · 2024-10-09
**Paper on relevant topic however more experimentation are needed**

**Confidence:** 4

**Summary:**

The paper addresses  what is the impact of reducing input features to predict sea ice concentration (SIC) and seeking some explainability on how the  features impact the prediction.  The paper presents a short description of the problem they authors addressed. In my understating, the paper does not present a new technique/approach to solve a problem: it only tries to improve previous results by  using different combinations  of the input features. This could be interesting for people in the climate-related field.

**Strengths:**

Tackle a relevant topic.
Good accuracy in the predictions.

**Weaknesses:**

Lack of novelty.
Lack of technical details: What are the bases to define the reduced and minimal feature setups (Sec. 3).
Lack of information on why the topic is important, what are the current issues, and how the authors's contribution provide a solution to the problem.

**Justification:**

More experimentation is required.
More details on the kind of problem they try to solve must be provided.
What is the relevance of the work within the field.

---

> ### Author Rebuttal · Authors · 2024-10-25
>
> First of all, we would like to thank the reviewer for his comment on the work. We revised the paper and hope that the adjusted text brings more clarity.
>
>
> The basis to choose the feature configurations is the previous work on feature importance by Joakimsen et al. [1], who use gradient saliency maps to assign the features importance scores. In their work, Joakimsen et al. conclude that mainly the features corresponding to the reduced configuration contribute to the prediction. By restricting the features to the minimal configuration, we extend the study to see how the model reacts to a very radical feature reduction. It represents a natural continuation of the feature reduction and is also based on the feature importance scores provided by Joakimsen et al. [1]
>
>
> We consider the topic of our paper relevant because it addresses the problem that convolutional neural networks often demand for high computational costs and require substantial storage capacity [2]. This paper provides a radical and novel approach compared to typical pruning methods, which focus on individual nodes, edges, etc. and not on the reduction of the input features [3]. In the introduction of our revised paper, we state this point more clearly.
>
>
> [1] H. L. Joakimsen, I. Martinsen, L. T. Luppino, A. McDonald, S. Hosking, and R. Jenssen. “Interrogating Sea Ice Predictability With Gradients”. In: IEEE Geoscience and Remote Sensing Letters 21 (2024), pp. 1–5. doi: 10.1109/ LGRS.2024.3366308.
>
>
> [2] Y. Cheng, D. Wang, P. Zhou, and T. Zhang. “Model compression and acceleration for deep neural networks: The principles, progress, and challenges”. In: IEEE Signal Processing Mag- azine 35.1 (2018), pp. 126–136.
>
>
> [3] P. Molchanov, A. Mallya, S. Tyree, I. Frosio, and J. Kautz. “Importance estimation for neu- ral network pruning”. In: Proceedings of the IEEE/CVF conference on computer vision and pattern recognition. 2019, pp. 11264–11272.

---

### Official Review · Reviewer_hzxt · 2024-10-15
**Investigating the Impact of Feature Reduction for Deep Learning-based Seasonal Sea Ice Forecasting**

**Confidence:** 5

**Summary:**

This work presents a novel model applied to climate forecasting. They reduced the number of features and preserve the performance of the model. Results exhibit how the features reduction improve the model in several cases.

**Strengths:**

Results exhibit how the features reduction improve the model in several cases.

**Weaknesses:**

Authors claim that "Our results showed that the models with fewer features generally provide higher accuracies in forecasts for all lead times" in the conclusion. Perhaps, it is not completely true as is evident in the results for extreme events. I recommend change this kind of sentences. Authors could also explain how the features reduction impact the performance in terms of computing.

**Justification:**

Results exhibit how the features reduction improve the model in several cases.

---

> ### Author Rebuttal · Authors · 2024-10-25
>
> We thank the reviewer for his comments.
>
> In the revised paper, we rephrased the way we conclude our results accordingly.
> We agree that a study of the computational performance would complement this work in a valuable way. However, due to a lack of time in this rebuttal period, we are not able to provide this analysis in the current paper but hope to investigate it in future work.

---

### Official Review · Reviewer_Grt6 · 2024-10-15
**Study of IceNet models under saliency-based feature reduction**

**Confidence:** 4

**Summary:**

This work investigates the impact of XAI methods in conjunction with IceNet (Andersson et al. 2021), a UNet model for forecasting sea-ice concentration (SIC) levels in the arctic. In particular, the work looks to investigate the impact of saliency based occlusion at the feature level to study the robustness and model performance under limitations of a reduced feature space. The authors base their approach on a gradient based saliency method developed explicitly for the purpose of SIC (Joakimsen et al. 2022), which in turn is largely based on early gradient based saliency via backpropagation (Simonyan et al. 2013). The manuscript makes a strong case for the adaptability of reduced feature models in improving forecast accuracy without compromising on performance, even in scenarios involving anomalous events.

**Strengths:**

- S1:  The study is well motivated, and tackles a pivotal modern problem -- accurate Arctic sea ice forecasting -- which is central to further advances in climate dynamics. The work could potentially have significant societal implications for further research in climate change mitigation and policy making.
- S2: The results are succinctly presented, and the ensuing discussion goes "the extra mile" to offer insightful observations, highlighting expected outcomes and anomalies. The discussion provides the reader with an understanding of the dynamics at play for the studied models.
- S3: The experimental setup and methodology is detailed, and the method seems reasonable and well motivated, grounded in important previous works in the field. In particular, this reviewer considered the proposed ensembling a nice touch; however, we would have liked to see even more investigations into the robustness of the model with uncertainty metrics for the reported results, which would likely strengthen the paper.
- S4: The approach shows clear benefits and practical applications for modelling techniques with explainability and interpretability methods, paving the way for further investigations into the effect of saliency based feature selection in sea-ice concentration forecasts. The use of saliency for feature reduction serve as an example of novel ways XAI methods can be exploited for modelling in interdisciplinary scientific studies.

**Weaknesses:**

- W1: The basis for saliency underlying the approach by Simonyan et al., and subsequently also for Joakimsen et al. has been shown to be more or less causally independent on the predictions of the model under scrutiny (Adebayo et al. 2018). In this work, the authors demonstrate that the saliency maps produced by gradient based methods demonstrate little change when replacing most of the layers of a convolutional network with randomly initialized weights. While this can be said to be a weakness of the aforementioned works, the reliance on this approach in the current work is also affected by proxy. This reviewer would have liked to seen applications of methods that are more robust in this regard, e.g., GradCAM or occlusion based methods, such as SHAP or LIME, particularly given the occlusion based approach of the study.
- W2: Table 1 shows accuracies over what is presumably the full ensemble of 10 models. Given the ensemble, is should be relatively straightforward to estimate the uncertainty of these predictions. This would, in this reviewers opinion, improve the presentation of the results, and provide the reader with an insight into the stability, reliability, and overall significance of the reported results.
- W3: The work seems to be heavily based on previous works, and could be presented with more clear delineation as a separate study, particularly in the abstract and introduction. On the first read through, the reader is left wondering as to the exact contributions of the current work. While the work "goes further" than the previous work, the extent of the scope of the current work is not too clear, particularly for researchers outside the niche of SIC forecasting. As it stands, the work is to be seen as an incremental study, as opposed to a significant step forward.
- W4: Given the nature of the study, it would be reasonable to ask for a robustness analysis of the model under more challenging perturbations. Given that convolutional models generally tend to be sensitive to perturbations, having estimates of the robustness of the trained models would ensure the reader that the results are not due to anomalies.

**Final Rebuttal Confidence:**

4

**Final Rebuttal Justification:**

The authors improved an already well formulated work with the feedback from the review. Our score still stands, and we recommend to accept the paper to the conference.

**Justification:**

The study reveals that the trained IceNet models are able to produce better results with lower lead times under the "reduced" feature space, as well as producing better results with longer lead times under the "minimal" feature space, and the authors discusses the implication for linear trend forecasting (LTF) for sea-ice concentration levels. Moreover, the study finds that the full model is still preferable in cases that exhibit higher deviations. As these feature spaces are reduced using the XAI method following Jacobsen et al., the authors demonstrate that this method has practical advantages in continued studies on sea-ice concentration forecasting, as well as interpretable modelling.

While this reviewer has some concerns about the specific methodology of saliency mappings under the approach outlined by Simonyan et al., we hope that this review can serve as motivation for the authors to dive deeper into the robustness (as well as statistical rigour) of the proposed method in future work.

As it stands, while the work must be seen as largely incremental, the work comes across as thorough while tackling an important problem. This reviewer therefore recommends the paper be accepted to the conference.

---

> ### Author Rebuttal · Authors · 2024-10-25
>
> We thank the reviewer for his valuable comments. We have revised the paper accordingly.
>
> W1: We agree that the consideration of other methods to supplement the saliency maps would indeed increase the robustness of the results for feature importance. As such studies were not feasible within the rebuttal period, we leave this open for future work. However, we acknowledge the limitations of the saliency maps in our revised paper in the “Related Work” section.
>
> W2: In the revised paper, the table was substituted by a figure. We address the uncertainty of the ensembles in the appendix.
>
> W3: We adjusted our introduction to show more clearly why our study should be seen as new and separate from previous work.
>
> W4: We could not provide a meaningful robustness study due to a lack of time in this rebuttal period but we included this as an outlook for future work.

---

### Official Review · Reviewer_ffrQ · 2024-10-16
**Review of Investigating the Impact of Feature Reduction for Deep Learning-based Seasonal Sea Ice Forecasting**

**Confidence:** 5

**Summary:**

The present paper performs an investigation of feature reduction in the traditional sea ice forecasting method. Considering the state-of-the-art IceNet model, a U-Net architecture for forecasting the Artic region sea ice with 50 features, the authors proposed two new sets of features to apply the same model: one with 21 features and another with just 11 features. They showed that the reduced features had good performance, better than the original set of features when forecasting normal events, but when we consider anomalous events, the use of the original set of features had better performance.

**Strengths:**

The authors combined the state-of-the-art IceNet model with the work of Joakimsen et al. to select only a few features for sea ice forecasting. They provided two sets of features, selecting which features Joakimsen et al. provided as important using X-AI. Using the three sets of features, they performed three different experiments: the first focusing on the anomalous month of September 2013, the second on all test data, and the third on the 10% most anomalous data. This is important to show how the feature selection behaves in relation to different difficulties in the data. The reduced set of features was good on general data but could not extrapolate for anomalous data, which could only forecast better results when considering all features.

**Weaknesses:**

The experiments could be extended using an ablation study to better determine which of the 50 features would help forecasting, complementing the X-AI approach from Joakimsen et al.

I want to understand the output of the U-Net model. You have the image size (432x432), 6 lead times, and 3 SIC classes? If the percentage can determine the SIC class, why would you need this new 3-value dimension? Please provide a text explaining the model.

As the authors showed a comparison between different feature sets, it would be interesting to see the difference in execution time between the experiments, showing the importance of using reduced feature sets in the forecasting.

**Justification:**

The submitted paper extends the work of Andersson et al. and Joakimsen et al. by experimenting with different sets of features for forecasting sea ice. They performed forecasting with different objectives, showing the strengths and weaknesses of the proposed feature sets for forecasting anomalous and well-behaved data.

As they provided a good set of experiments to show their proposed approach, I recommend it to be accepted.

---

> ### Author Rebuttal · Authors · 2024-10-25
>
> We thank the reviewer for sharing his interesting comments and suggestions to this work. We want to reply to the comments as follows.
>
> Weakness 1: Extending experiments with an ablation study.
>
> Answer:
> We agree that the feature importance estimate could be designed more robust by considering different methods apart from gradient-based saliency maps. We acknowledge this in the “Related Work” section of the revised work. Including such a complementary study in this work is due to shortage of time and space unfortunately not possible.
>
>
> Weakness 2: Output of the IceNet model.
>
> Answer:
> As the reviewer’s questions correctly suggests, the output of the IceNet model consists of predictions for six lead times. Each of these six predictions itself comprises of three image-like (432x432 pixel) grids, one for each SIC class. The values that build up these grids correspond to the probabilities of the pixels to fall in the respective SIC class.
> For clarity it should be stated that the classes themselves are defined by the percentage values of the SIC. However, these percentages are distinct from the probability values that for the class affiliations in the model output.
>
> For the accuracy evaluation, classes 2 and 3 are combined to a single class “ice” (as opposed to “no ice” which is represented by class 1), i.e. probabilities of the classes are summed. Using these results, a binary prediction (“ice” or “no ice”) is produced according to the higher probability value for the two outcomes. This is done, as the threshold of 15 % from class 1 is a common metric in many sea ice applications. Doing so, we also follow the previous IceNet evaluations and provide better comparability.
>
> We edited our explanation of the model in the revised paper and hope this will bring more clarity.
>
>
> Weakness 3: Execution time differences
>
> Answer:
> We agree that a comparison of execution time would naturally complement this study and be interesting to investigate. However, due to a lack of time in this rebuttal period, we are not able to provide this analysis in the current paper but hope to investigate it in future work. As we used “early stopping “ in our model training, it is also hard to give an estimate of the actual calculate time differences.

---

### Meta-Review · Area_Chair_6yVn · 2024-10-31

**Recommendation:** Accept (Poster)
**Confidence:** 4

**Metareview:**

Quality:
The paper is methodologically sound, grounded in significant prior work, and provides detailed experimental setups and analyses. It introduces a comprehensive evaluation of feature reduction in sea ice forecasting using the IceNet model, examining reduced feature sets in normal and anomalous data scenarios. However, suggestions for improvement include an ablation study to determine specific feature contributions and uncertainty quantification for ensemble results.

Pros:
- Effective reduction in the feature space, achieving high accuracy on general data while maintaining performance on anomalous events.
- Clear motivation and presentation of experiments with thoughtful observations on expected versus observed outcomes.
- Practical implications for climate change modeling and policy-making in the Arctic.

Cons:
- The reliance on saliency-based methods from earlier work (Joakimsen et al., Simonyan et al.) has limitations, as gradient-based saliency maps may lack causality with model predictions.
- The study could benefit from using more robust XAI techniques (e.g., GradCAM, SHAP, or LIME) and a robustness analysis under perturbations.
- The incremental nature of the study could have been clarified more, especially for readers outside this niche field.

Clarity:
- The work is well-written with clear motivations and structured analyses, but the introduction and abstract could benefit from a clearer distinction of the novel contributions.

Originality:
- The work is novel in its application of feature reduction in Arctic sea ice forecasting using XAI-driven methods. However, it heavily builds on previous studies, which may make the contributions seem incremental.

Significance:
- The research is of moderate significance, primarily valuable for the climate modeling and forecasting community. The exploration of feature reduction in the IceNet model holds practical implications, but further robustness analyses could enhance its broader applicability.

**Suggested Changes To The Recommendation:**

3: I agree that the recommendation could be moved up

---

### Decision · Program_Chairs · 2024-11-06

**Decision:**

Accept (Oral)

**Comment:**

Given the AC positive recommendation, we recommend an oral and a poster presentation given the AC and reviewers recommendations.